



**The SPARC water vapor assessment II: intercomparison of satellite and ground-based microwave measurements**

Gerald E. Nedoluha[1], Michael Kiefer[2], Stefan Lossow[2], R. Michael Gomez[1], Niklaus Kämpfer[3],
Martin Lainer[3], Peter Forkman[4], Ole Martin Christensen[4], Jung Jin Oh[5], Paul Hartogh[6], John
Anderson[7], Klaus Bramstedt[8], Bianca M. Dinelli[9], Maya Garcia-Comas[10], Mark Hervig[11], Donal
Murtagh[4], Piera Raspollini[12], William G. Read[13], Karen Rosenlof[14], Gabriele P. Stiller[2], Kaley A.
Walker[15]

[1]Remote Sensing Division, Naval Research Laboratory, Washington, D.C., USA

[2]Karlsruhe Institute of Technology, Institute of Meteorology and Climate Research, Karlsruhe,
Germany

[3]Institute of Applied Physics, University of Bern, Bern, Switzerland

[4]Onsala Space Observatory, Department of Radio and Space Science, Chalmers University of
Technology, Onsala, Sweden

[5]Sookmyung Women's University, Seoul, South Korea

[6]Max Planck Institute for Solar System Research, Katlenburg-Lindau, Germany

[7]Hampton University, Hampton, Virginia, USA

[8]University of Bremen, Institute of Environmental Physics, Bremen, Germany

[9]Istituto di Scienze dell'Atmosfera e del Clima del Consiglio Nazionale delle Ricerche, Bologna,
Italy

[10] Instituto de Astrofisica de Andalucia, CSIC, Granada, Spain

[11]GATS Inc., Driggs, Idaho, USA

[12]Istituto di Fisica Applicata 'Nello Carrara' (IFAC) del Consiglio Nazionale delle Ricerche
(CNR), Firenze, Italy

[13]Jet Propulsion Laboratory, California Institute of Technology, Pasadena, California, USA

[14]University of Colorado, Atmospheric Chemistry Observations & Modeling Laboratory,
Boulder, Colorado, USA

[15]University of Toronto, Department of Physics, Toronto, Ontario, Canada

*Correspondence to:* Gerald E. Nedoluha (nedoluha@nrl.navy.mil)



**Abstract**

As part of the second SPARC (Stratosphere-troposphere Processes And their Role in Climate)
water vapour assessment (WAVAS-II), we present measurements taken from, or coincident with,

seven sites from which ground-based microwave instruments measure water vapor in the middle
atmosphere. Six of the ground-based instruments are part of the Network for the Detection of
Atmospheric Composition Change (NDACC) and provide datasets which can be used for drift
and trend assessment. We compare measurements from these ground-based instruments with
satellite datasets that have provided retrievals of water vapor in the lower mesosphere over

extended periods since 1996.

We first compare biases between the satellite and ground-based instruments from the upper
stratosphere to the upper mesosphere. We then show a number of time series comparisons at
0.46 hPa, a level that is sensitive to changes in $H_2O$ and $CH_4$ entering the stratosphere, but,
because almost all $CH_4$ has been oxidized, is relatively insensitive to dynamical variations.

Interannual variations and drifts are investigated both with respect to the Aura Microwave Limb
Sounder (MLS) (from 2004 onwards), and with respect to each instrument's climatological
mean. We find that the variation in the interannual difference in the mean $H_2O$ measured by any
two instruments is typically ~1%. Most of the datasets start in, or after, 2004, and show annual
increases in $H_2O$ of 0-1%/year. In particular, MLS shows a trend of between 0.5%/year and

0.7%/year at the comparison sites. However the two longest measurement datasets used here,
with measurements back to 1996, show a much smaller trend of between +0.1%/year and -
0.1%/year.

## 1. Introduction

Since the early 1990's ground-based microwave (GBMW) instruments have been measuring

profiles of water vapor in the middle atmosphere for the detection of long-term change. These
ground-based measurements were used in the mid-1990's for satellite intercomparison studies
with instruments on the Upper Atmosphere Research Satellite (UARS), as well as with shuttle-
borne instruments during the Atmospheric Laboratory for Application and Science (ATLAS)
missions (Nedoluha et al., 1997). Longer term comparisons were made with measurements from

the HALOE instrument (e.g. Nedoluha et al., 2003), and several GBMW instruments were





included in the first SPARC water vapor assessment (Kley et al., 2000). A survey of water vapor intercomparisons through 2010 is given by Hocke et al. (2013). More recently, there have been multi-year comparisons between GBMW instruments and Aura MLS and MIPAS (Nedoluha et al., 2013a).

The retrieval of water vapor from GBMW instruments makes use of high spectral resolution measurements of emission near the 22.235 GHz rotational transition of water vapor. The retrieval of water vapor profiles as a function of altitude (or pressure) from the measured spectrum relies upon the sensitivity of the emission at each altitude to pressure broadening. These measurements are taken nearly continuously, and, depending primarily upon the altitude to

which the retrieval is desired, generally require from several hours to a week of measurement integration. Retrievals are physically possible from the stratosphere to the upper mesosphere (the latter requiring the longest integration periods), however long-term stability is difficult to maintain below the upper stratosphere because of the difficulty of maintaining a stable instrumental spectral baseline. The optimal retrieval levels for long-term ground-based

measurements are therefore in the lower mesosphere.

While all of the ground-based measurements shown here are with instruments measuring at 22.235 GHz, the measurements come from different groups, each of which have developed and deployed their own instruments. Each group has its own retrieval code, but all of them make use of the optimal estimation technique (Rodgers, 1976). A detailed explanation of the general

retrieval technique and its application to microwave radiometry is given in Kämpfer et al. (2013).

We will first present profile comparisons between satellite and GBMW instruments at a number of sites based upon averages of coincident measurements. We will then take advantage of the fact that ground-based instruments provide the longest available datasets of $H_2O$ from the upper stratosphere to the upper mesosphere, and are therefore especially useful for studies of the

stability of satellite measurements in these regions over extended periods of several years. We will then make use of these ground-based measurements to assess the stability of the instruments making these measurements, both satellite and ground-based, and to assess the long-term variations of $H_2O$.



## 2. The measurement datasets

In this study we show water vapor measurements at seven sites where GBMW instruments have been deployed. For these sites we show measurements from the ground-based instruments and from satellite instruments which make measurements near those ground-based sites. For the six
5 sites for which we have data sets covering at least four years we will show time series and investigate temporal variations.

### 2.1 The ground-based microwave radiometer datasets

We will present measurements from seven GBMW instruments. Six of these instruments are
10 part of the Network for the Detection of Atmospheric Composition Change (NDACC) and provide datasets which can be used for drift and trend assessment. Although there is a stronger water vapor emission line at 183 GHz that could be used for observations under dry conditions from high altitude sites, this emission line is too optically thick to be useful for most locations. Hence, all of these instruments observe the 22 GHz emission line, which allows nearly
15 continuous observation from most sites.

The 22 GHz radiometer at Onsala Space Observatory (57°N, 12°E) was built and is operated by Chalmers University of Technology in Gothenburg (Forkman et al., 2003). The data are available since 2002, and cover the vertical range ~45-80 km with a measurement response (Connor et al., 1991) of >0.75. The receiver consists of a heterodyne receiver based on an
20 uncooled high electron mobility transistor (HEMT) preamplifier, while the back end is based upon a digital FFT (Fast Fourier-Transform) spectrometer with a bandwidth of 200 MHz over 16000 channels.

The MIddle Atmospheric WAter vapour RAdiometer (MIAWARA) was built in 2002 at the Institute of Applied Physics (University of Bern) and has been continuously operating on the
25 roof of the building for Atmospheric Remote Sensing in Zimmerwald close to Bern (46.7°N, 7°E) since September 2006. The vertical resolution of the instrument varies between 11 km in the stratosphere and 14 km in the mesosphere. A former measurement range from approximately 7 – 0.1 hPa (Deuber et al., 2005) was extended to roughly 10 – 0.02 hPa with instrumental upgrades in spring 2007. An Acousto-Optical Spectrometer (AOS) was replaced by a digital FFT
30 spectrometer, which improved the spectral resolution from 600 kHz to 61 kHz.





The Seoul WAter vapor RAdiometer (SWARA) was developed, like MIAWARA, at the Institute of Applied Physics at the University of Bern and has been operational since October 2006 at the Sookmyung Women's University of Seoul (37.3°N, 126°E) in South Korea (De Wachter et al., 2011). SWARA is in principle a copy of MIAWARA and the same specifications apply.

However, as the wings of the spectrum are affected by baseline ripples, the retrieval bandwidth is limited to 50 MHz, and the measurement response is <0.60 at altitudes below ~38 km (~4 hPa).

There are currently three Water Vapor Millimeter-wave Spectrometer (WVMS) instruments taking science quality data (a fourth WVMS instrument is currently being used to help develop a new feedhorn). The instruments were developed and built at the Naval Research Laboratory

with funding from NASA, and are operating at Table Mountain, California (34°N, 242°E), Mauna Loa, Hawaii (19.5°N, 204°E), and Lauder, New Zealand (45°S, 170°E). Early WVMS measurements from Table Mountain are described in Nedoluha et al. (1995), and the evolution of the WVMS systems is described in Gomez et al. (2012). The original instruments at each of these sites have all been replaced by fourth-generation instruments.

Much of the development work on the fourth-generation instruments was done at the JPL Table Mountain site, and the WVMS4 has now been taking science quality measurements since 2010. A previous system (WVMS2) operated at this site from 1992 until 1997 and measured increasing water vapor in the early 1990's (Nedoluha et al, 1998), however these measurements will not be shown here. Measurements at Mauna Loa were taken by the WVMS3 system starting in March

1996. The new WVMS6 system at Mauna Loa has been taking measurements since 2011. Similarly, at Lauder the WVMS1 system, which had been taking measurements since November 1992, was replaced with the WVMS7 instrument in November 2011. Here we will only use data from Lauder back to the beginning of 1996. While the new instruments use FFT's and measure over a spectral width of 500 MHz, the retrievals for this study make use only of the same 60

MHz spectral width used by the older instruments.

The cWASPAM (cooled Wasserdampf- und Spurengasmessungen in der Atmosphäre mit Mikrowellen) instrument (Hallgren and Hartogh, 2012) has performed observations at ALOMAR (Arctic Lidar Observatory for Middle Atmosphere Research, 69°N, 16°E) in Northern Norway since the summer of 2008. The data included here cover 2008 to 2011. It replaced an

older instrument that took measurements at the same location since 1995 (Hartogh and Jarchow,



1995). The instrument was developed by the Max Planck Institute for Solar System Research (now located in Göttingen) and is characterized by high sensitivity. This was achieved by cooling of the horn antenna and the hot load (including the amplifier and cold load). The signal detection is performed by two Chirp Transform Spectrometers, analyzing both the vertical and

horizontal polarization of the signal. The spectrometers have a bandwidth of 40 MHz which is divided into 4096 channels with an effective resolution of 10 kHz. Overall this instrument allows us to obtain water vapor information from ~40 km to 85 km.

The measurements from Onsala, Bern, Seoul, Mauna Loa, and Lauder, were all compared with Aura MLS in Haefele et al. (2009). Comparisons were made between 0.03 hPa and 3 hPa, with

the comparisons clearly degrading below 1 hPa. For the case of the Seoul measurements the seasonal comparisons were only performed at 0.01 hPa and 0.03 hPa. Only the Mauna Loa and Lauder measurements performed well in the upper stratosphere (Haefele et al., 2009). There have since been efforts to extend the useful lower altitude of some of these measurements (Nedoluha et al., 2011; Lainer et al., 2015)

**2.2 Satellite datasets**

We make use here only of satellite datasets which provide information in the lower mesosphere, which were operational for a period since 1996, and for which there are coincidences with at least one of the seven ground-based measurement sites. This leaves us with measurements from ACE-FTS, HALOE, MIPAS, Aura MLS, SCIAMACHY, SMR, and SOFIE. In many cases we

will show results from multiple retrievals from these instruments. For ACE-FTS we show the two most recent retrieval sets (v2.2 and v3.5; Boone et al., 2005; 2013) while those for Aura MLS are from v4.2 (https://mls.jpl.nasa.gov/data/v4-2_data_quality_document.pdf). The HALOE comparisons are with the v19 retrieval set, and we will restrict ourselves here to measurements since 1996. For SCIAMACHY we show the solar occultation Optimal Estimation

Method (OEM) retrievals which are sensitive in the upper stratosphere and mesosphere. For SMR we show retrievals from the 489 GHz and strong 557 GHz bands which are both sensitive to $H_2O$ in this region of the atmosphere. Both the 489 GHz and the 557 GHz SMR observations have been available since late 2001 and there are typically four measurement days monthly. The 557 GHz SMR observations (Lossow et al., 2007) do not cover the stratosphere and are therefore

not included in any of the other SPARC water vapor assessment studies.



From 2002 to 2004 MIPAS operated in a single high spectral resolution mode. In 2005 the
MIPAS measurement scheme underwent a fundamental change from a high-resolution mode to a
reduced spectral resolution mode. For time series comparisons we will not make use of the high-
resolution mode retrievals, but these are included in the bias comparisons when coincidences are

available. From 2005 onwards, in addition to a nominal mode, there were a number of special
measurement modes. A daily list of the measurement mode for MIPAS is provided at
http://eodg.atm.ox.ac.uk/MIPAS/L1B/ . Here we will make use of two types of measurement
modes; nominal (NOM), in which the lowest altitude measurements are taken in the troposphere
and middle atmosphere (MA), in which the lowest altitude measurements are in the lower

stratosphere, but extend into the thermosphere. MIPAS measurements are processed by four
different processing centers: 1) The University of Bologna (Dinelli et al., 2010); 2) The
European Space Agency (ESA) (Raspollini et al., 2013), IMK/IAA (von Clarmann et al., 2009;
Stiller et al., 2012; Garcia-Comas et al., 2016), and Oxford (Payne et al., 2007)), and all of these
retrievals are available for public use. Since certain datasets may be best suited to specific

science applications, we include a number of these datasets in this comparison. The four
processors differ in their choices of spectral ranges (so called micro-windows) used, the vertical
grid on which the retrievals are performed (pressure or geometric altitude), the choice of
regularization (and related to this, the vertical resolution), the choice of spectroscopic data base,
the sophistication of the radiative transfer (in particular, whether or not non-LTE emission is

considered), whether or not any attempt is made to account for horizontal inhomogeneities, etc..
Some of the different processing schemes also make use of different level-1b data versions (here
V5 and V7) based on different ESA calibrations. The spread of results seen for MIPAS
indicates how specific choices within a retrieval approach may influence the retrieval results.

The list of retrieval datasets to be included, and the color-coding which we will use throughout

this study, is given in Figure 1. Further details relevant to all of the satellite datasets are given in
Walker et al. (in preparation).

## 3. Average Profile Comparisons

Retrieved profiles from GBMW measurements generally have a vertical resolution of ~10-15km.

Because this is significantly coarser than the typical vertical resolution of the limb scanning



satellite measurements used here, the satellite retrievals should generally be convolved before being compared with ground-based retrievals. Thus, instead of comparing the retrieved ground-based profile with the vertical profile $x_{sat}$, one calculates a convolved satellite profile ($x_{sat\_conv}$) by applying the equation: $x_{sat\_conv} = x_a + \mathbf{A} (x_{sat} - x_a)$, where $x_a$ is the a priori mixing ratio profile and

$\mathbf{A}$ is the averaging kernel calculated from the microwave measurement. The application of this equation not only helps to address the problem of differences in vertical resolution, but also ensures that, at altitudes where the microwave measurement is insensitive, both the retrieved microwave profile and the convolved satellite profile are equal to the a priori.

A typical set of averaging kernels for the GBMW retrievals used in this study is shown in Figure

2. Ideally the sum of the kernels for a particular level is ~1.0, with lower values indicating increased a priori dependence. As is apparent in the figure, the measurement sensitivity of the GBMW retrievals decreases and the vertical resolution degrades with increasing altitude in the upper mesosphere, and the retrieval becomes increasingly dependent upon the a priori mixing ratio profile.

While GBMW retrievals are generally provided over a fixed pressure (or altitude) range, a useful bias comparison should minimize the effect of a priori information in the retrievals. The GBMW profiles which are used for bias comparisons are therefore all required to have averaging kernels for which the sum of the kernels is at least 0.5 for pressures from 0.03 hPa to 3 hPa. We note that the sum of the kernels is a slightly different (but more tractable) measure of sensitivity than

measurement response. Tropospheric opacity due to weather conditions can affect the temporal resolution required to achieve a desired sensitivity, and in particularly humid conditions useful GBMW measurements may not be possible. Summer months tend to have a wetter troposphere, degrading the microwave profile measurements; hence there is a tendency for more comparisons to take place during winter months. This is particularly the case for the satellite comparisons

with GBMW measurements at ALOMAR and Seoul.

The equation for the convolution of satellite measurements may require profile information that is outside the altitude range provided by the measuring instrument. This applies here specifically to those satellite retrievals which do not extend into the upper mesosphere. To allow for the application of averaging kernels over their full range, the satellite profiles, where necessary, are

extended beyond their standard retrieval range. Above the highest valid satellite retrieval



altitude a climatological profile is used, which is scaled to the topmost valid satellite measurement point. In order to minimize the effect of this extension on the comparisons, we include only measurements at altitudes at least 10 km below the topmost measurement altitude for that particular profile. We then calculate the value $< vmr_{sat} - vmr_{GBMW} > / < vmr_{GBMW} >$ at

each altitude based on all of the profiles that reach a particular altitude (once the top 10 km has been removed). This necessarily results in a different number of comparisons at each level. We then show comparisons only at levels where, using the criterion above, at least 50% of the satellite measurements are available; hence the highest comparison altitudes will be 10km below the altitude where 50% of the satellite measurements are valid.

In Figure 3 we show for a number of satellite retrievals the average difference relative to all available ground-based microwave measurements. These differences are calculated from all coincidences for which there is a satellite measurement that is spatially within 1000 km and within +/-5° latitude of the ground-based site, and that is made either within the integration time range of the GBMW instrument, or within +/-24 hours of the center of this integration period (for

integrations shorter than 48 hours). All of the satellite measurements shown in Figure 3 have been convolved with averaging kernels from the appropriate GBMW instrument.

As can be seen in Figure 3, the GBMW retrievals are generally slightly lower than those from most satellites over most of the vertical range shown. In comparisons with Aura MLS, this difference is almost everywhere within 10%, indicating good agreement in the shape of the

vertical profile. The only levels where this difference exceeds 10% is at the highest altitudes, but we note that the mixing ratios are decreasing rapidly with increasing altitude in this region. This is especially true for the ALOMAR comparison, which take place preferentially in the winter when mesospheric water vapor is especially low. Only the measurements from ALOMAR show a difference with respect to MLS which goes outside the 0-10% range at pressures below 0.05

hPa. The GBMW comparisons with ACE-FTS are very similar to those with MLS, except that the ACE retrievals are 0-5% lower. Comparisons with HALOE are only available for the three GBMW that were operational in 2005 (Lauder, Mauna Loa, and Onsala). These GBMW retrievals show higher mixing ratios (up to ~10%) than HALOE except near the top altitudes of the comparisons. Thus, the GBMW retrieved $H_2O$ mixing ratios are, almost everywhere, larger

than those from HALOE, but smaller than those from MLS.





All of the MIPAS retrievals shown in Figure 3 are from the version 5 (V5) Level-1B spectra. As mentioned in 2.2, the high resolution (V5H) MIPAS measurements were only available until 2004. The IMK/IAA high resolution retrievals show mixing ratios that are larger than the GBMW values everywhere, with a maximum difference relative to the three GBMW sites of

~10-20% near ~1 hPa, and minimum differences of ~5% at ~0.3 hPa (the top of the retrieval range for this satellite measurement). The Bologna high resolution retrievals are quite similar to the IMK/IAA retrievals, but ~2% lower. The ESA high resolution retrievals are generally similar to the IMK/IAA retrievals from 3 to 1 hPa, but then drop slightly more rapidly with decreasing pressure. They are available to a slightly lower pressure level than the IMK/IAA

retrievals, and at this lowest pressure level (~0.3 hPa) they show mixing ratios ~10-15% lower those retrieved from the GBMW measurements. Conversely, the Oxford high resolution retrievals, which show mixing ratios similar to the other two high resolution retrieval versions up to ~0.5 hPa, show higher mixing ratios at the top few levels, and are 10-15% higher than the GBMW at ~0.3 hPa.

The NOM MIPAS retrievals, taken in the reduced spectral resolution measurement mode since 2005, are generally available up to ~0.2 hPa. With the exception of the comparisons near ALOMAR, they all show mixing ratios higher than those from the GBMW retrievals at pressure levels from 3 hPa to ~0.6 hPa. The largest difference in this pressure range is at Lauder, where the MIPAS NOM retrievals are up to ~20% larger near ~2 hPa. The ESA retrievals tend to give

the highest mixing ratios, although for comparisons near Bern the MIPAS retrievals are highest at the lowest pressure levels. Between ~0.6 hPa and ~0.2 hPa the GBMW and MIPAS NOM retrievals are almost always with ~10%. The IMK/IAA, ESA, and Oxford comparisons show a minimum relative to the GBMW values at the lowest pressure level at all sites except ALOMAR.

The MA MIPAS retrievals are intended for studies at higher altitudes than the NOM retrievals.

The MA retrievals from ESA and Bologna are shown in Figure 3 to have very similar averages to the NOM version, but the ESA MA retrieval does go to a slightly higher altitude. The IMK/IAA and Oxford MA retrievals cover the entire pressure range shown. The IMK/IAA MA retrievals show decreasing mixing ratios with increasing altitude relative to the GBMW instruments (with the exception of ALOMAR) from approximately the stratopause to 0.1 hPa,

and tend to be approximately constant at higher altitudes. The Oxford MA retrievals similarly





show a decrease relative to the GBMW retrievals with increasing altitude over much of the mesosphere, but this decrease tends to be much more gradual after starting at a higher altitude. As a result, at ~0.2 hPa the Oxford retrievals generally show mixing ratios ~5-10% larger than those from the GBMW retrievals, while the IMK/IAA retrievals are generally ~5% lower than

the GBMW retrievals.

Comparisons with the 557 GHz SMR retrieval cover the top of the altitude range shown in Figure 3, and those with the 489 GHz retrieval the bottom. Both SMR retrievals are generally 0-10% lower than the GBMW retrievals, with the 557 GHz retrieval somewhat larger in the region of overlap. The top of the 489 GHz retrieval (~0.16 hPa) is 20-30% lower than the GBMW

retrieval in all but one of the comparisons. SOFIE comparisons are only possible at the high latitude (69°N) ALOMAR site, where they show mixing ratios that are, at most levels, lower than any instrument except SMR. Similar to the other satellite comparisons at this site the SOFIE retrievals show a minimum with respect to the GBMW near ~2 hPa and near ~0.07 hPa. SCIAMACHY comparisons are only possible with the three northernmost sites, i.e. Bern,

Onsala, and ALOMAR, and only at the highest pressure levels (there is only one level of overlap with Onsala). The differences are ~5% at Bern, ~13% at Onsala, and varying between ~0-16% at ALOMAR, where the variation with pressure matches that of most other satellite-based retrievals in this pressure range.

## 4. Relative Instrumental Drifts

In this Section we will examine temporal variations in the six $H_2O$ datasets with at least 4 years of data. In order to allow for a detailed study of these variations we will focus on the 0.46 hPa pressure surface. There are both instrumental and geophysical reasons for focusing this study of temporal change in the lower mesosphere.

Retrievals from GBMW instruments can provide information from the mid-stratosphere to the

upper mesosphere. However, as mentioned in Section 1, the stability of the ground-based $H_2O$ measurement datasets degrades with decreasing altitude in the stratosphere. In the upper mesosphere the $H_2O$ emission becomes weaker with increasing altitude; hence the retrievals become increasingly dependent upon the a priori (Figure 2). Retrievals in this region require increasingly long integration periods to achieve a given measurement sensitivity. The best



altitude region for ground-based microwave measurements to study long-term changes in $H_2O$ is therefore the lower mesosphere. Fortunately, the lower mesosphere is geophysically also an ideal region for the study of long-term changes in $H_2O$.

In the stratosphere $H_2O$ increases with altitude as $CH_4$ is oxidized (Le Texier et al., 1988;
Wrotny et al., 2010). As this oxidation occurs gradually, the amount of $H_2O$ that has been produced by this process in the stratosphere depends upon the age of the parcel, and this is affected by variations in dynamics. This sensitivity to dynamics-driven changes shows up in the amplitude of the seasonal cycle, and some latitudes (particularly from the tropics to SH midlatitudes) have a large annual cycle in the upper stratosphere (cf. Lossow et al., 2017). Once
air has reached the lower mesosphere, however, almost all of the $CH_4$ has been oxidized, hence these chemical-dynamical variations no longer change the amount of $H_2O$ in an air parcel.

Studies in the upper mesosphere are complicated by an increasingly large seasonal cycle (especially in percentage terms) with increasing altitude, especially at higher latitudes (Lossow et al., 2017; Figure 1). In addition, in the upper mesosphere variations in Lyman-α radiation cause
variations in the photodissociation of $H_2O$, and this causes both solar-cycle-driven decadal-scale changes (Nedoluha et al., 2009) and diurnal changes (Scheiben et al., 2013) in $H_2O$ , both of which increase rapidly with increasing altitude.

In the lower mesosphere, however, interannual variations of $H_2O$ reflect primarily the changes in $H_2O$ and $CH_4$ entering the stratosphere. As shown in Figure 2.2 of IPCC Chapter 2 (Hartmann,
et al., 2013) the increase in $CH_4$ has been ~50 ppbv since the mid-1990s, which, once fully oxidized, would result in a gradual increase in $H_2O$ of ~0.1 ppmv, or ~1-2%, over the past 2 decades. Interannual variations in $H_2O$ in the lower mesosphere on shorter timescales than this must be attributed to other physical mechanisms, such as variations in $H_2O$ entering the lower stratosphere, and it has been suggested that the changes observed in $H_2O$ in the lower
mesosphere from 2004-2013 were not inconsistent with the effects of changes in tropical tropopause temperatures (Nedoluha et al., 2013b).

For comparisons in the lower mesosphere we choose coincidence criteria based on the measurements (both satellite and ground-based) and the geophysical properties of this region. Since, unlike for the overall bias comparisons, we require coincident measurements over a





number of separate time intervals, we use a coarser set of coincidence criteria than were used in Section 3. For most instruments we use a latitudinal coincidence criterion of +/-5º and longitudinal coincidence of +/-30°. If we calculate the standard deviation of the differences between any two sets of coincident measurements (using coincidence criteria of +/-3.5 days and

+/-5° latitude) we find that σ values are generally within the range 0.3 to 0.7 ppmv (4-10%). Assuming 52 weeks of coincident measurements in a year, this would result in a formal 2σ error of up to ~0.2 ppmv (~3%) for an annual average. The standard deviation of the differences remains very similar whether or not one imposes a longitude coincidence criterion or uses a zonal average.

For the sparser solar occultation measurement datasets (HALOE, ACE, and SCIAMACHY) we do not impose any longitudinal coincidence criteria, but use the zonal average of measurements within +/-5° latitude. We also extend the temporal coincidence to +/-7 days. Under nominal operation conditions HALOE and ACE typically measure near a mid-latitude site ~10-15 times per year, hence the formal 2σ error for an annual average might be as large as ~0.4 ppmv (~6%).

Aura MLS has been providing nearly global daily coverage of $H_2O$ in the middle atmosphere since August 2004 and thus provides an ideal dataset to which the other measurements can be compared during this entire period. We first compare these measurements to the six available ground-based measurements, and to other available satellite measurements coincident with these ground-based stations, in order to evaluate the consistency with which variations in $H_2O$ are

tracked by the different instruments. All of the comparisons are based on annual averages measured at 0.46 hPa. Note that while the ground-based sites do cover a range of latitudes, the latitude range is by no means complete and there is only one site in the Southern Hemisphere.

Unlike the comparisons in Section 3, none of the satellite data shown here has been convolved with averaging kernels from the ground-based instruments. We did convolve the entire Aura

MLS dataset with a typical set of ground-based microwave averaging kernels, and found that at the 0.46 hPa level shown here the differences were nearly indistinguishable. This insensitivity to convolution is because, in this altitude region, neither the water vapor profile nor the anomalies in the profile change rapidly with altitude, and the ground-based retrievals are only weakly dependent upon the a priori mixing ratio profile.



We have added, here and in Section 5, one additional retrieval dataset. This is the MIPAS-ESA V7R NOM dataset. The MIPAS V7 retrievals differ from the MIPAS V5 retrievals in that they use the level 1b radiances version 7, where a time-dependent non-linearity correction scheme has been adopted to account for the change in non-linearities over the course of the mission due to

aging of the detectors (Valeri et al., 2017). This correction introduces an altitude-dependent temporal change in the MIPAS retrievals. The difference in the temporal variations between the MIPAS-ESA V7R NOM and MIPAS ESA V5R NOM datasets can be taken as representative of temporal variations between any of the MIPAS V7 and V5 retrievals at 0.46 hPa.

In Figure 4 we show the percentage difference between the annual average for coincident

measurements (both satellite and ground-based) with MLS, i.e.

$$f_{inst}(t) = 100 \left( vmr_{inst} - vmr_{MLS} \right) / vmr_{MLS}$$ averaged over all measurements for a full year. The colors and symbols, which are those shown in Figure 1, are based upon the instrument being compared with MLS. These annual averages are shown 4-times per year, covering approximately Jan.-Dec., Apr.-Mar., Jul.-Jun., and Oct.-Sept., with, e.g., the Jan.-Dec. 2005

average being plotted at 2005.5. Each measurement is therefore included in four of the anomaly datapoints shown in the figure. The 489 GHz SMR data are both lower than most of the other measurements, and show a strong positive trend; hence, while there are data throughout this time period, many of the measurements towards the beginning of the time period do not appear in Figure 4.

Figure 4 shows that, at 0.46 hPa, annual average mixing ratios measured by GBMW instruments are, with a few temporary exceptions between ~0% to 10% lower than MLS (as in Figure 3). An interesting point to note is that, of the six GBMW instruments, five of them have a lower mixing ratio relative to MLS at the end of the timeseries than at the beginning (although in the case of Table Mountain this difference is only ~1%). While there is a drop in the GBMW measurements

relative to MLS, the precise timing of this downward drift relative to MLS is not the same for all of these instruments. The drop in GBMW mixing ratio relative to MLS at Mauna Loa occurs primarily from 2005 to 2008, the drop at Bern occurs from 2007 to 2009, and the drop at Lauder occurs from 2008 to 2011. Although most of the GBMW measurements do show an overall negative trend relative to MLS, perhaps the most important conclusion that can be drawn from



these comparisons is that there is no particular period during which a preponderance of measurements show a clear increase or decrease relative to MLS.

The sign of the drift between the GBMW and MLS measurements is consistent with that reported by Hurst et al. (2016), which found that frost point hygrometer measurements at four of

five sites showed a negative drift relative to MLS. However, the results shown in Hurst et al. (2016) indicate that this drift began around 2010. From 2010 to 2014 the Lauder, Mauna Loa, Table Mountain, and Bern instruments are all stable relative to MLS. The GBMW instrument at Seoul does show a very large drop relative to MLS from 2010 to 2012, while for the GBMW instrument at Onsala we only have data to the end of 2012, pending a reprocessing of the dataset.

Just as for the GBMW-MLS intercomparisons, most of the satellite retrievals are also lower than, but within 10% of MLS, during most of the comparison period. Exceptions are the SCIAMACHY, the MIPAS Oxford MA, and the MIPAS ESA NOM retrievals, which are almost always higher than MLS, and the 489 GHz SMR retrievals, which are almost always more than 10% lower.

Figure 5 provides some statistical measures of the relative stability of the measurement datasets shown in Figure 4. In calculating the results for Figure 5 we use the calendar year average differences from Figure 4 and simply fit a 2-term linear trend, so that $f_{inst}(t) = A_0 + A_1 t$ . The drift, as measured by the linear trend ($A_1$) term, thus provides an estimate of the relative stability of trends for each dataset relative to MLS over the entire period of the instrument comparison. The

x-axis in Figure 5 shows the variation in the mean absolute interannual difference from the calendar year averages, i.e. $x_{inst} = <|f_{inst}(t) - <f_{inst}(t)>|>$. This gives an indication of the accuracy to which it is possible to measure year-to-year variability.

In addition to showing these statistics specifically for the temporally coincident measurements, we have also included in Figure 5 smaller symbols which show annual average differences

between MLS and a comparison instrument for all measurements taken near a site over the same time years. The annual averages are calculated as follows. First, we calculate an instrument specific climatology $C_{inst}(t)$ for all measurements coincident with each site by fitting each dataset with a 5-parameter fit

$$C_{inst}(t) = A_1 + A_2 \sin(2\pi t) + A_3 \cos(2\pi t) + A_4 \sin(4\pi t) + A_5 \cos(4\pi t) \quad (1)$$



where $t$ is in years. We then subtract $C_{instrument}$ ($t$) from the measurements for that year and calculate an average annual anomaly. The subtraction of the seasonal fit from the data should reduce the effects of year-to-year seasonal variations in the sampling.

We include both types of comparisons because we would like to assess the uncertainty in the
annual average $H_2O$ variations. In cases where both the MLS and the comparison measurement are available nearly continuously the two methods should give nearly the same result, and this is the case. However, if one uses this method for temporally limited samples, such as for SCIAMACHY measurements near Bern (a latitude which SCIAMACHY reaches only from mid-May to mid-July) then the best-fit to (1) can result in unphysical annual cycles, so the relative
drift analysis is best done with coincident comparisons. SCIAMACHY measurements are also made near Onsala, and here, while there is an offset between the two methods, the variations calculated by the two methods are similar. In general, comparisons of annual data are most difficult at higher latitudes, both because of the larger annual cycles, and because sudden warmings can result in rapid descent which cause rapid changes in $H_2O$ mixing ratio at a
particular altitude (cf. Straub et al., 2012). In cases such as this annual averages can be significantly affected by whether or not measurements are taken during a particular period.

In addition to the comparisons with MLS we also include on Figure 5, using the HALOE colors from Figure 1, comparisons between HALOE and the GBMW at Mauna Loa and Lauder. The mean interannual differences between the HALOE-GBMW pairs (shown in HALOE colors) are
clearly much larger than between the MLS-GBMW pairs (shown in GBMW colors) at these sites. To some extent this may be because calculated HALOE trends are necessarily based on sparser sampling than MLS, but we also note that significant steps have been taken to improve the stability of the GBMW instruments at these two sites since the 1990s (Gomez et al., 2012), so the better MLS-GBMW agreement is probably, at least to some extent, a result of improved
GBMW stability for the Lauder and Mauna Loa systems.

Figure 5 shows that, from one year to the next, the difference between the annual average $H_2O$ measured by MLS and by one of the other instruments included in this study, using either comparison method, is ~1%. ~34% of the comparisons above show a mean absolute interannual difference of <1%, and ~48% show a difference of <1.2%. Based on the MLS data, the





interannual variation of the geophysical mean for these six sites over the period during which MLS has been making measurements is ~1.4%.

With the exception of the GBMW instrument at Onsala, all of the GBMW instruments show a negative drift relative to MLS. Four of the instruments (Lauder, Mauna Loa, Bern, and Table

Mountain) show drifts that are ~-0.5%/yr. At Mauna Loa, Bern, and Table Mountain the drifts relative to MLS of the GBMW instruments are quite similar to those of the MIPAS-ESA V7 NOM retrieval, whereas at the Southern Hemisphere Lauder site almost all of the MIPAS retrievals show a more positive drift.

## 5. Measured Changes in Water Vapor

From the similarity of the large and small symbols in Figure 5 we conclude that, generally, comparisons of coincident measurements produce drifts and interannual differences that are similar to those calculated from anomalies relative to instrument-specific climatologies. Thus, the anomalies relative to instrument-specific climatologies give useful estimates of interannual variations. Having reached this conclusion, we shall proceed to show interannual variations

relative to such instrument-specific climatologies.

In Figure 6 we show these annual anomalies plus the constant term ($A_1$) from the 5-parameter fit. We start the time series in 1996, since this is the first year for which at least two of the ground-based measurement datasets are available. Figure 6 allows us to investigate not just variations relative to MLS, but geophysical variations as observed by each instrument.

The only instruments measuring during much of the 1996-2004 period are the GBMW instruments at Mauna Loa and Lauder, and HALOE. The GBMW measurements from Lauder from 1996-2004 show more interannual variability than HALOE, but the overall change from the beginning to the end of this time period is small. The difference between measurements taken during calendar years 1996 and 2004 is +0.08 ppmv for the GBMW and -0.17 ppmv for

HALOE. The GBMW measurements from Mauna Loa show much less interannual variation, and also only a small overall change from 1996-2004. The difference between the annual average from the GBMW measurement from July 1996 to June 1997 (i.e. the first full year of measurements) and those for the 2004 calendar year is -0.11 ppmv, while for the HALOE measurements over this period the difference is -0.17 ppmv. Measurements from the GBMW at





Onsala become available in 2002. In agreement with the 557 GHz SMR measurements these show a large decrease from 2002 to 2003 (-0.6 ppmv in the GBMW and -0.5 ppmv in the SMR measurements), but this change is not observed in the HALOE and SCIAMACHY datasets, and only to a smaller extent in the 489 GHz SMR dataset.

There was a decrease in water vapor measured by many instruments at 0.46 hPa between 2005 and 2006. The calendar year 2006 MLS measurement anomalies at all six sites were lower than those in 2005 by from -0.05 ppmv at Lauder to -0.17 ppmv at Onsala. 76 of the 89 (85%) retrieval sets showed a decrease in the annual anomalies between 2005 and 2006. From 2006 onwards there has been an increase in $H_2O$ at these altitudes, as was shown by Nedoluha et al.
(2013b). They noted very good agreement between the increase observed by GBMW measurements from Mauna Loa and global measurements from MLS and MIPAS when comparing annual averages for 2006 and 2011.

While there seems to have been a general decrease in 0.46 hPa water vapor between 2005 and 2006, the opposite occurred between 2007 and 2008. If we compare the calendar year 2008 and
2007 measurement anomalies from MLS we find that there was an increase at all sites except Lauder (i.e. all of the Northern Hemisphere sites). The increases ranged from between +0.05 ppmv at Mauna Loa to +0.16 ppmv at Bern. 71 of the 76 (93%) retrieval datasets showed an increase in annual anomalies between 2007 and 2008.

One apparent feature in Figure 6, is that the annual average $H_2O$ mixing ratio for almost all of
the retrievals is almost always larger than that measured by HALOE at any time. This helps to emphasize that the interannual, or even decadal-scale variations in $H_2O$ are generally smaller than the absolute differences between instruments. Any understanding of long-term changes in $H_2O$ would therefore be irreparably harmed by measurements gaps. Such gaps would eliminate the critical period of comparison of coincident measurements to understand absolute differences.

In Figure 7 we show linear trends derived from each full dataset, thus the calculated trends cover many different time periods, although the period between 2004 and 2012 is especially well represented. Since trends, especially those calculated from the shorter timeseries, may be affected by the phase of the QBO during which measurements start or end, we use in this case an 8-parameter fit



$$C_{inst}(t) = A_1 + A_2 \sin(2\pi t) + A_3 \cos(2\pi t) + A_4 \sin(4\pi t) + A_5 \cos(4\pi t) +$$

$$A_6 \, QBO30(t) + A_7 \, QBO50(t) + A_8 \, t \qquad (2)$$

where QBO30 and QBO50 are the 30 hPa and 50 hPa zonal wind anomalies from the Climate
Prediction Center (www.cpc.ncep.noaa.gov/data/indices).

Figure 7 shows that most retrievals indicate an increase in $H_2O$, with the MLS measurements
showing an increase of ~0.5%/year at all sites. The MIPAS measurements cover a similar, but
slightly shorter period to MLS, and while most of these retrievals show positive trends, the single
MIPAS V7 retrieval shown here shows a negative trend at the Northern Hemisphere sites. The
HALOE measurements, because we have limited this study to measurements from the beginning

of 1996 onwards, show a negative trend. However, we note that if we include the full HALOE
dataset back to 1991, then the overall trend is between +0.1%/year and +0.6%/year at all sites.
The two longest datasets, the GBMW measurements from Lauder and Mauna Loa, show very
little change since 1996. The difference between the first and last annual averages shown in
Figure 6 for these two instruments is +0.27 ppmv at Lauder and +0.13 ppmv at Mauna Loa. This

compares to an expected increase in $H_2O$ from $CH_4$ oxidation over this time period of ~+0.1
ppmv, and hence implies that there has been very little increase in $H_2O$ entering the stratosphere
since 1996. However, the Lauder and Mauna Loa GBMW instruments show a negative trend of
~-0.5%/yr relative to the MLS measurements. If this relative change indicates a problem with
the GBMW measurements over the past decade then this would add ~+0.35 ppmv to the change

in $H_2O$ mixing ratio, an increase that clearly would imply an increase in $H_2O$ entering the
stratosphere.

## 6. Summary

We compared satellite and GBMW measurements at a number of sites. We began with profile
comparisons at seven sites from 3 hPa to 0.03 hPa. Comparisons between satellite and GBMW

measurements over this range of pressures generally showed agreement within 10%, with most
satellite retrievals showing altitude varying differences resulting in lower mixing ratios than the
GBMW retrievals at some levels, and higher mixing ratios at others. The exception to this rule
was the GBMW-MLS comparisons, which had vertical profiles with very similar shapes. As a



result, at six of the seven sites, these comparisons showed that the MLS measured water vapor was at almost all levels between 0% and 10% higher than the GBMW measured water vapor.

Temporal variations on annual scales were studied at six NDACC sites. This analysis was limited to the 0.46 hPa level, a level which is ideal for the study of $H_2O$ trends for both

geophysical and instrumental reasons. We compared the interannual variation all of the available measurements with MLS at the six sites, and found, using two different intercomparison methods, that the relative variation between MLS and other measurement datasets in the annual average was typically ~1%. We did find that four of the GBMW instruments showed trends relative to MLS of ~-0.5%/year, but we noted that there were differences in the detailed temporal

evolution of that drift. At Mauna Loa, Bern, and Table Mountain the drifts of the GBMW instruments relative to MLS are quite similar to those of the MIPAS-ESA V7 NOM retrieval, whereas at the Southern Hemisphere Lauder site almost all of the MIPAS retrievals show a more positive drift. We also found that, at all sites, MLS showed an increase of ~0.6%/year.

We also compared trends in $H_2O$ at all of the sites over the measurement time period which was

available for each instrument. While the preponderance of retrievals showed an increasing trend of 0-1%/year in 0.46 hPa $H_2O$, the retrievals were particularly concentrated over the 2004-2012 period. However longest retrieval datasets (GBMW from Lauder and Mauna Loa) showed trends of between +0.1%/year and -0.1%/year from 1996-2015.

### 7. Acknowledgments

Work at the Naval Research Laboratory was funded by NASA under the Upper Atmosphere Research Program and by the Office of Naval Research. We want to express our gratitude to SPARC and WCRP (World Climate Research Programme) for their guidance, sponsorship and support of the WAVAS-II programme. The Atmospheric Chemistry Experiment (ACE), also known as SCISAT, is a Canadian-led mission mainly supported by the Canadian Space Agency

and the Natural Sciences and Engineering Research Council of Canada.





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



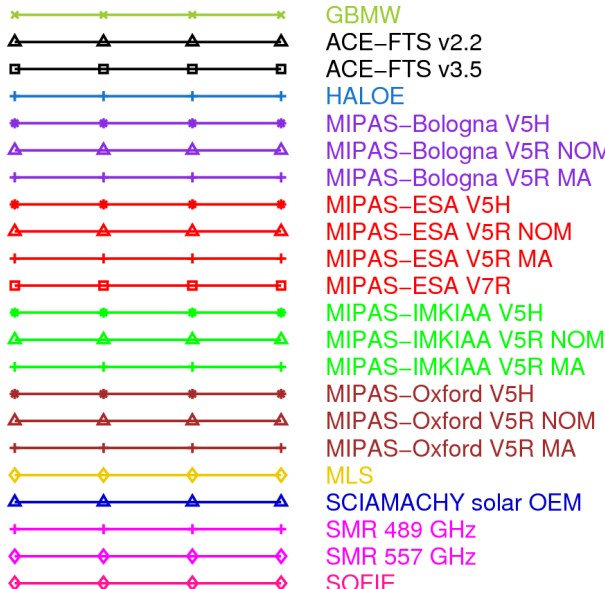

Figure 1 - The color and symbol scheme used for instruments and their retrieval versions used in comparisons throughout this study.





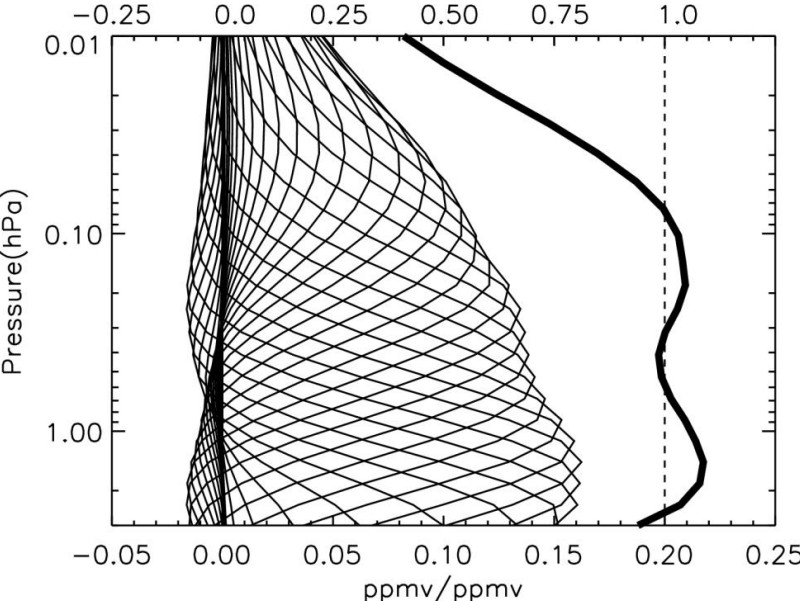

Figure 2 – A typical set of averaging kernels for GBMW retrievals (in this case for retrievals from Mauna Loa). The thin lines are referenced to the lower x-axis and represent the sensitivity of the measurement to perturbations at individual pressure levels. The single thicker line is referenced to the upper x-axis indicates the sum of the averaging kernels at that level. Ideally the sum of these kernels is near unity, as indicated by the dashed line.





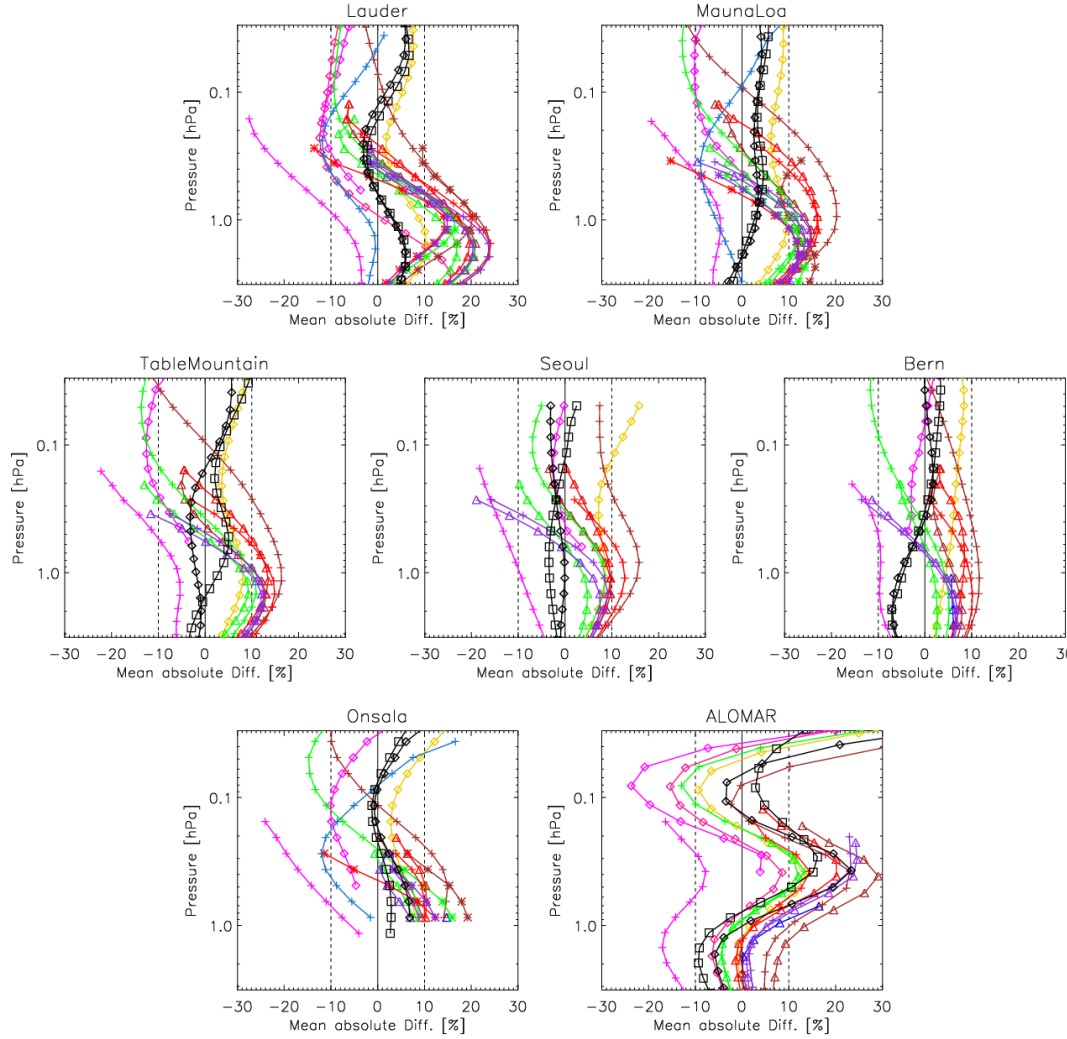

Figure 3 – The average difference between coincident measurements at seven ground-based sites from 3 hPa to 0.03 hPa, shown from South to North. The difference shown is convolved satellite minus GBMW using the satellite symbols given in Figure 1.



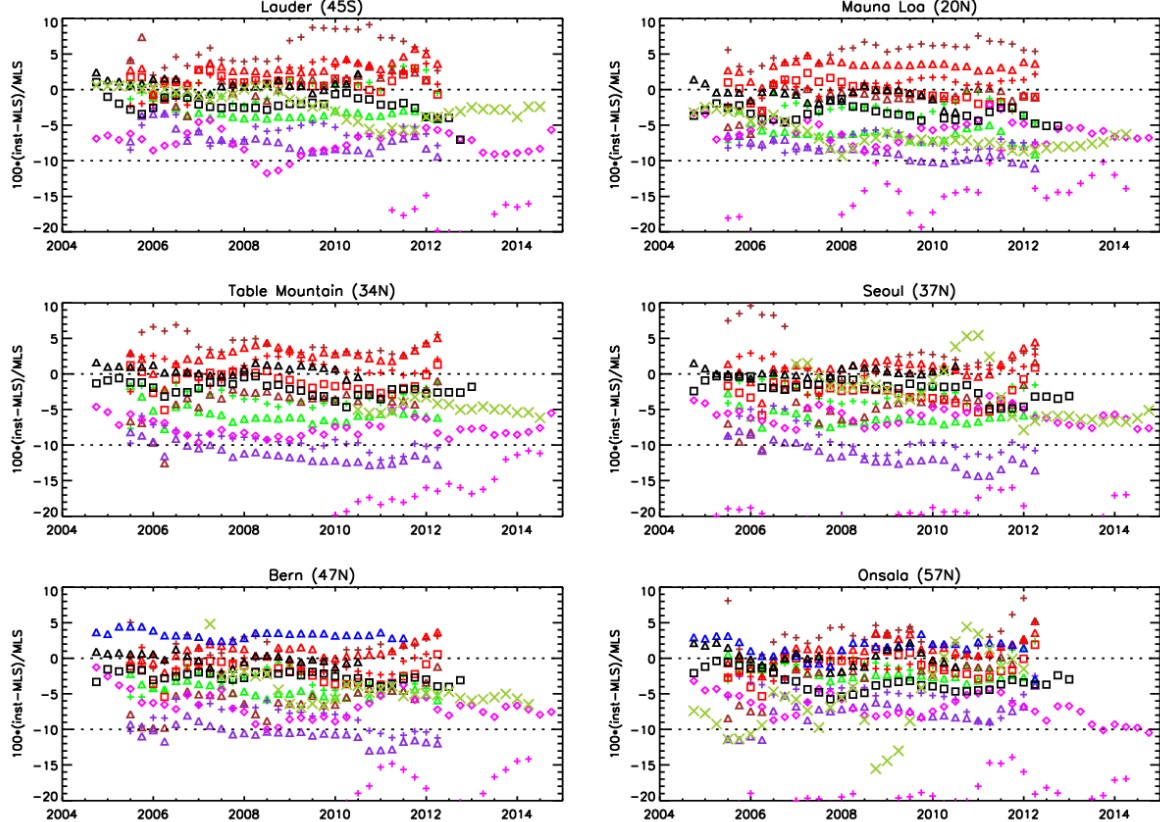

Figure 4- Annual average differences between coincident $H_2O$ measurements at 0.46 hPa.
Results are shown at six ground-based sites and all differences are with respect to MLS
measurements at those sites. Annual average differences are shown 4-times per year (see text).
The symbols used are from Figure 1, and indicate the instrument which is being compared with
MLS.





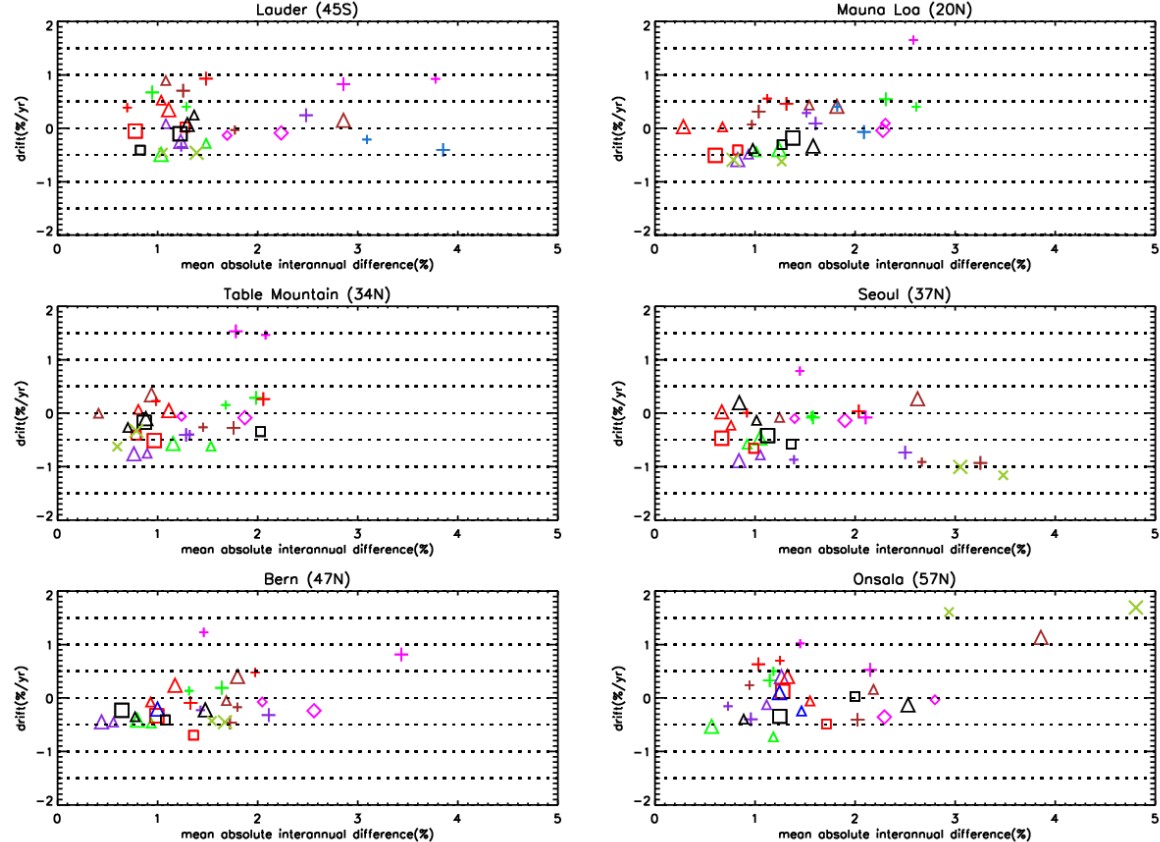

Figure 5– The drift (y-axis) and mean absolute interannual difference (x-axis) between
coincident H$_2$O measurements at 0.46 hPa.  Results are shown at six ground-based sites and,
unless otherwise indicated, all differences are with respect to MLS measurements at those sites.
The large symbols indicate differences for coincident comparisons, as were shown in Figure 2.
The small symbols indicate comparisons of annual average differences from climatologies (see
text for details).   The symbols used are from Figure 1, and indicate the instrument that is being
compared with MLS.  The overlap period between HALOE and MLS is too short for these
analyses, but we do perform these analyses for GBMW vs. HALOE comparisons at Lauder and
Mauna Loa.  These are indicated using the HALOE symbols from Figure 1.





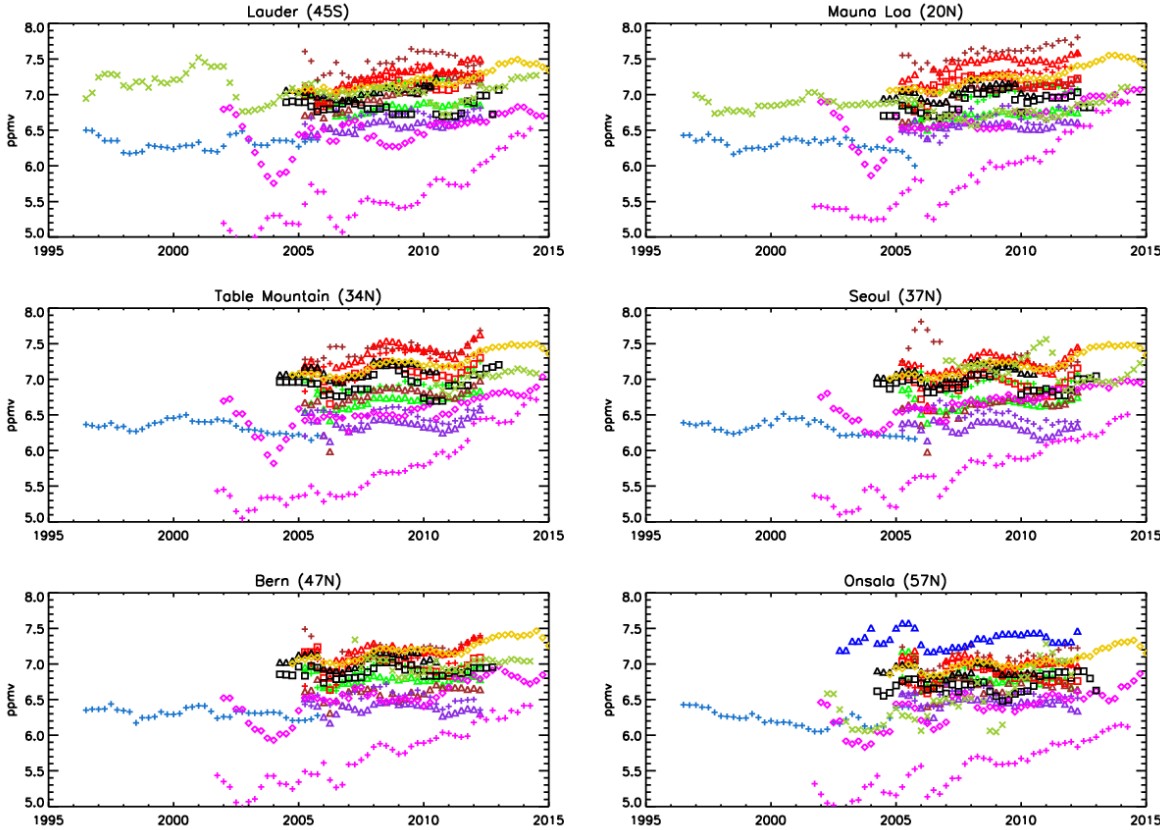

Figure 6 – Measurements since 1996 at 0.46 hPa from, or coincident with, six NDACC sites. The results shown are the annual anomaly plus the constant term from the 5-parameter fit (see text). Values are shown 4-times per year. Symbols and colors are from Figure 1.




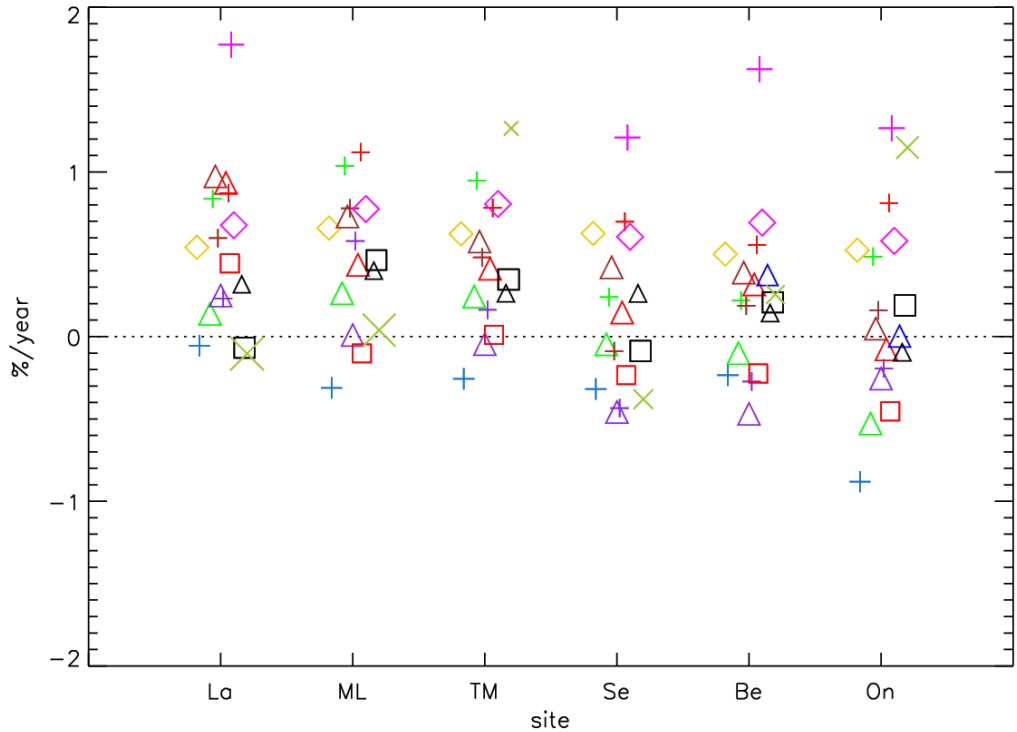

Figure 7– Linear trends at 0.46 hPa at each of the six NDACC ground-based sites. Trends are calculated over the data taking period for each instrument shown in Figure 6. The sites are listed from South to North: Lauder (45°S), Mauna Loa (20°N), Table Mountain (34°N), Seoul (37°N), Bern (47°N), and Onsala (57°N). Symbols are slightly offset from each other along the x-axis for legibility. Larger symbols indicate longer datasets.