# Peer review of "The SPARC water vapor assessment II: intercomparison of satellite and ground-based microwave measurements"

_Atmospheric Chemistry and Physics, 2017_

## Referee Comment (RC1) · H. C. Pumphrey (Referee) · 8 Aug 2017

**1 General comments**

This paper is well written, adequately illustrated, and forms a clear and useful summary of the material covered. The analysis appears to have been done carefully and diligently. It should therefore be published subject to minor corrections, most of which are technical.

[Figure]

**2 Specific comments**

- Page 6 Line 22: It is not clear why the authors felt it necessary to show two different versions of the ACE-FTS data and only one version of the MLS data. I have no reason to suggest that they change this, but I think they should give reasons for their choice.

- The political point made by the authors on page 18 line 22 is very pertinent. It would be nice (but perhaps inappropriate) if they were to be more explicit about the fact that both ACE-FTS and MLS are near the end of their lives and that there are no missions any further on than a drawing board which might continue the time series.

**3 Technical corrections**

- All figures except figure 2: The scheme of colours and symbols set out in figure 1 has a number of unsatisfactory features. In particular, there are too many red/magenta colours which are not easy to distinguish. This makes all the figures (but especially figure 5) rather hard to interpret.

- Figure 2: The reader has no way to tell which curve is for the sensitivity at which pressure level. The curves should really use different line-styles or colours, with a legend to show which curve is for which level.

- Figure 5: the many dotted horizontal lines are a distraction. The authors should consider removing all apart from -1, 0 and 1, and making the line at 0 dashed rather than dotted.

- Page 15 line 18 and page 17 line 16: A$_1$ should be $A_1$, i.e. the "A" should be in italic.

- Page 16 line 1: $C_{instrument}$ should be $C_{inst}$ for consistency with Eq. 1.
* * *

---

## Referee Comment (RC2) · Anonymous Referee #2 · 22 Aug 2017

This manuscript is an important manuscript for the assessment of water vapor in the upper stratosphere and mesosphere. The authors compare several ground based microwave radiometer observations with a number of satellite retrievals from several different instruments. This paper has good scientific merit; however, the presentation quality needs to be improved. For that reason I recommend this paper for publication after major revisions to give the authors sufficient time to deal with the recommendations, which I describe in detail below.

Major comments

The paper shows water vapor data from 22 different retrievals from 8 instruments.

[Figure]

Combining all of these into single figures makes these very difficult to read. Furthermore, 13 of these retrievals are from the MIPAS instrument, i.e. different analyses of raw data from the same instrument. This heavily biases the appearance of the figures towards MIPAS and covers the data from other instruments, which have only one or two retrieval versions. One possible solution for this problem could be to discuss all MIPAS retrievals separately and show only one MIPAS retrieval in the comparison with the other instruments. This would significantly improve the clarity of the figures and the overview of all available instruments. Throughout the paper it has been very difficult to follow the discussion of individual instruments, when data are hidden in the cloud of all other instruments or difficult to distinguish from similar colors and symbols used for other instruments.

In the presentation of the time series there is no discussion of the changes in the GBMW instruments throughout the time series. There have been a number of changes in the instruments, but no discussion about the possible impact of these changes on the time series. Page 5, Lines 13ff indicate that the WVMS instruments have been replaced, but there is no indication, when this happened, if there has been some overlap period, when both instruments measured, of if there has been any effort to evaluate the impact of this change on the time series. I would assume that the other instruments have had changes throughout time as well. Some discussion should take place to that effect to convince the readers that a possible impact on the time series is much less than the differences seen in the comparisons with the other instruments and the trend estimates discussed in the paper.

Specific comments

Why did the authors choose to show the two most recent ACE-FTS retrievals? It would appear that the recent version 3.5 would be sufficient.

Have any of the surface instruments ever been compared to each other or is there a traveling standard that has been shipped to the different sites?

Page 5, Line 18: Why are these older measurements not used here?

Page 5, Line 24-25: Why is the smaller spectral width used in this paper, when a wider range is available? It would be nice if the authors could give a short statement on the impact of this decision on the analysis.

Page 5, Line 6: What does a measurement response of "<0.6" mean?

Page 9, Line 21: It could be helpful, if the authors showed a typical water vapor profile over the entire upper stratosphere to mesosphere range.

Page 10, Line 1: Why are the MIPAS V7 data excluded here? They are included in the later discussion.

Page 10, 1st paragraph: The MIPAS data shown in Figure 3 justify a stand-alone figure to discuss the MIPAS specific differences. Can the authors make a statement, which of the many MIPAS version is preferable? After all, they are all from the same instrument.

Page 10: The other instruments should be separated from Figure 3, since these are very difficult to identify.

Page 13, Line 26: I would accept "nearly indistinguishable" if the authors could clarify that the differences are significantly less than 2%. If not, they become relevant in comparison to the annual average difference above. Please clarify.

Page 15, Line 3: The sign of the drift is coincident with Hurst et al. (2016). How about the magnitude?

Page 15, Line 29: Why do you believe this is a sufficient model, i.e. without a QBO term as in equation 2?

Page 20, last paragraph: Can this be interpreted such that there has been no significant trend in water vapor at the 0.46 hPa level and that the longest time series limit possible trends to within +/- 0.1%/year? This is just for clarification and could probably be elaborated more in a sentence or two.

Technical comments

Figure 1 should indicate which MLS version and which HALOE version are being used.

Page 13, Line 7: I guess you mean "∼0.2 ppmv (∼3%) for an annual average difference", not "annual average".

Page 17, Line 7: The conjunction "whereas" seems somewhat misplaced, since the two phrases do not connect well. A part of the explanation seems to be missing.

――――――――――――――――――

---

## Author Comment (AC1) · 12 Oct 2017

Thanks very much to Hugh Pumphrey for some good suggestions, especially for improvements to the figures. We repeat below the reviewer comments in and give our responses in italics.

Page 6 Line 22: It is not clear why the authors felt it necessary to show two different versions of the ACE-FTS data and only one version of the MLS data. I have no reason to suggest that they change this, but I think they should give reasons for their choice.

*We did drop the v2.2 retrievals from Figure 6 to reduce clutter, but since there are clear differences from the v3.5 retrievals, and since v2.2 retrievals have been used in previous published studies, we left them in Figures 3 and 4. We now state that the MLS v3.3 retrievals were almost imperceptibly different.*

• The political point made by the authors on page 18 line 22 is very pertinent. It would be nice (but perhaps inappropriate) if they were to be more explicit about the fact that both ACE-FTS and MLS are near the end of their lives and that there are no missions any further on than a drawing board which might continue the time series.

*We of course completely agree with the reviewer regarding this serious problem, but we are hesitant to mention it here. Such a comment would more appropriately be made in the SPARC WAVAS overview paper in preparation (Walker et al.).*

3 Technical corrections
• All figures except figure 2: The scheme of colours and symbols set out in figure 1 has a number of unsatisfactory features. In particular, there are too many red/magenta colours which are not easy to distinguish. This makes all the figures (but especially figure 5) rather hard to interpret.

*The color table in Figure 1 is being used in a number of studies, and for consistency with those studies we are very hesitant to make any changes. We very much apologize for the difficulty this causes in interpreting Figure 5.*

• Figure 2: The reader has no way to tell which curve is for the sensitivity at which pressure level. The curves should really use different line-styles or colours, with a legend to show which curve is for which level.

*Figure 2 now includes different colors for specific indicated levels.*

• Figure 5: the many dotted horizontal lines are a distraction. The authors should consider removing all apart from -1, 0 and 1, and making the line at 0 dashed rather than dotted.

*Done*

• Page 15 line 18 and page 17 line 16: A1 should be A1, i.e. the "A" should be in italic.

*Done. Sorry for being sloppy about this.*

---

## Author Comment (AC2) · 12 Oct 2017

Thanks very much to Reviewer 2, who asked many very reasonable questions that led us to make a number of changes (including to most of the figures) that we hope will clarify and improve the paper. We repeat below the reviewer comments below and give our responses in italics.

The paper shows water vapor data from 22 different retrievals from 8 instruments. Combining all of these into single figures makes these very difficult to read. Furthermore, 13 of these retrievals are from the MIPAS instrument, i.e. different analyses of raw data from the same instrument. This heavily biases the appearance of the figures towards MIPAS and covers the data from other instruments, which have only one or two retrieval versions. One possible solution for this problem could be to discuss all MIPAS retrievals separately and show only one MIPAS retrieval in the comparison with the other instruments. This would significantly improve the clarity of the figures and the overview of all available instruments. Throughout the paper it has been very difficult to follow the discussion of individual instruments, when data are hidden in the cloud of all other instruments or difficult to distinguish from similar colors and symbols used for other instruments.

*We certainly cannot disagree that there is a very large amount of information on each figure and that this sometimes makes it difficult to distinguish individual instruments. But first, in defense of the large number of MIPAS measurements shown, we should point that these are not just different retrievals using the same raw data, but, in many cases, analyses of different raw datasets. The high spectral resolution, reduced spectral resolution nominal, and middle atmosphere modes are truly distinct measurements. Even multiple retrievals of specific measurement modes may make use of different specific wavelengths.*

*With the exception of Figure 3, the MIPAS V5H retrievals are not presented (because they covered on 2 years), so this leaves nine MIPAS retrievals. In an effort to make things a bit more readable, and to deemphasize MIPAS, we have made the following changes:*
*1) We split Figure 3 into two separate figures. We have, as suggested by the reviewer, split these between MIPAS and non-MIPAS measurements.*
*2) We removed the 4 MIPAS-NOM retrievals from Figure 6. This leaves 5 MIPAS retrievals on this figure. We also removed the ACE-FTS v2.2 retrievals.*
*3) In order to deemphasize MIPAS we slightly reduced the MIPAS symbol sizes in Figures 4 and 6.*

In the presentation of the time series there is no discussion of the changes in the GBMW instruments throughout the time series. There have been a number of changes in the instruments, but no discussion about the possible impact of these changes on the time series. Page 5, Lines 13ff indicate that the WVMS instruments have been replaced, but there is no indication, when this happened, if there has been some overlap period, when both instruments measured, of if there has been any effort to evaluate the impact of this change on the time series. I would assume that the other instruments have had changes throughout time as well. Some discussion should take place to that effect to convince the readers that a possible impact on the time series is much less than the differences seen in the comparisons with the other instruments and the trend estimates discussed in the paper.

*We certainly understand the reviewer's concern on this point as this is an issue that members of the GBMW community must deal with very carefully. We have added the following paragraph in Section 2.1:*
*"Given the small number of GBMW instruments, the difficulty of moving them and ensuring consistency in the measurements, and the absence of any other ground-based technique which could be used as a travelling standard, the standard method of minimizing disruptions to GBMW timeseries is to compare with a well established satellite instrument before and after any major change. While this clearly invalidates the GBMW measurements as an independent standard during the period being used to ensure consistency, the major changes to GBMW instruments used in this study are sufficiently infrequent that they do not present an obstacle to multi-year assessments."*

Why did the authors choose to show the two most recent ACE-FTS retrievals? It would appear that the recent version 3.5 would be sufficient.

*We did drop the v2.2 retrievals from Figure 6 to reduce clutter, but since there are clear differences from the v3.5 retrievals, and there are published studies with the v2.2 retrievals, we left them in Figures 3 and 4.*

Have any of the surface instruments ever been compared to each other or is there a traveling standard that has been shipped to the different sites?

*We now address this issue in the paragraph added in 2.1 (mentioned above). Given the complexity of these instruments local comparisons are rare. There are currently 2 WVMS instruments at Mauna Loa, but, until very recently (2 months ago) one of them did not have a feedhorn antenna of sufficiently high quality to make comparisons worthwhile.*

Page 5, Line 18: Why are these older measurements not used here?

*When we deployed the newest instruments we found we had to recalibrate our new data (by the same amount at Lauder and Mauna Loa) so that it was consistent with our old dataset. This is the only time in 20+ years that we have ever had to do this, and we do not understand why it was necessary. We are still experimenting and trying to understand why this recalibration was necessary, but until we better understand the offset between the new and old instruments we think it is inappropriate to show the new and old Table Mountain GBMW measurements together, since any differences could be incorrectly interpreted in terms of a geophysical change.*

Page 5, Line 24-25: Why is the smaller spectral width used in this paper, when a wider range is available? It would be nice if the authors could give a short statement on the impact of this decision on the analysis.

*This is a very reasonable question, and we have added the following sentence to the text:*
*"The wider spectral bandwidth does provide profile information down to the mid-stratosphere, but we have found that the optimization of our retrieval over a larger spectral width can, given imperfectly characterized instrumental baseline changes, adversely affect the consistency of our mesospheric retrievals."*

Page 5, Line 6: What does a measurement response of "<0.6" mean?

*This phrase comes up earlier in 2.1 as well, where we have added to the sentence: "The data are available since 2002, and cover the vertical range ~45-80 km with a measurement response (Connor et al., 1991) of >0.75" the hopefully descriptive phrase: "(i.e. the a priori contribution to the retrieval is <0.25)." We also added another reference to (Connor et al. 1991) next to the "<0.6" mentioned here.*

Page 9, Line 21: It could be helpful, if the authors showed a typical water vapor profile over the entire upper stratosphere to mesosphere range.

*This has been added to Figure 2.*

Page 10, Line 1: Why are the MIPAS V7 data excluded here? They are included in the later discussion.

*Thanks for noticing this. The MIPAS V7 retrieval was added very late in the writing of the manuscript after this paragraph was written. It has now been added to Figure 3.*

Page 10, 1st paragraph: The MIPAS data shown in Figure 3 justify a stand-alone figure to discuss the MIPAS specific differences. Can the authors make a statement, which of the many MIPAS version is preferable? After all, they are all from the same instrument.

*As suggested, the MIPAS data are now shown as a stand-alone part of Figure 3. While we cannot conclude from our study that any particular MIPAS measurement is preferable, we do now include a lengthy paragraph in the Section 4 discussing this issue.*

Page 10: The other instruments should be separated from Figure 3, since these are very difficult to identify.

*We have split this figure into 2 parts.*

Page 13, Line 26: I would accept "nearly indistinguishable" if the authors could clarify that the differences are significantly less than 2%. If not, they become relevant in comparison to the annual average difference above. Please clarify.

*This is a really good point. It is true that on the figure indicated they are nearly indistinguishable, but we have added to Figure 5 a symbol showing the difference between MLS and convolved MLS variations. Thanks for this idea.*

Page 15, Line 3: The sign of the drift is coincident with Hurst et al. (2016). How about the magnitude?

*We have added the numbers from Hurst et al., but it is not clear how to make an appropriate comparison in magnitude given the different length time periods.*

Page 15, Line 29: Why do you believe this is a sufficient model, i.e. without a QBO term as in equation 2?

*Equation (2) contains a trend term, so to calculate an accurate trend we wanted to do our best to limit the effect of the QBO on the trend calculation. But here the idea is to compare interannual differences, so if the instruments show different sensitivities to the QBO we would like this to show up in the plot. Of course one can include a QBO fit (and perhaps even a trend term) here, and then just use the annual terms to calculate the anomalies. But the 5-parameters in (1) are only very slightly different compared to those obtained from a full 7 (or 8) parameter fit; hence the effect on Figure 5 is hardly noticeable. We felt that the additional explanation of using a 7 (or 8) parameter fit and then only using the first 5 terms would be confusing.*

Page 20, last paragraph: Can this be interpreted such that there has been no significant trend in water vapor at the 0.46 hPa level and that the longest time series limit possible trends to within +/- 0.1%/year? This is just for clarification and could probably be elaborated more in a sentence or two.

*No, we certainly did not wish to imply that the 20 year trend is known to within +/-0.1%/year, but as written this could be the interpretation. The last two sentences of this paragraph have been rewritten. We also rewrote the last sentence of the abstract in order to, hopefully, not give this impression.*